# Chronic and Cycling Hypoxia: Drivers of Cancer Chronic Inflammation through HIF-1 and NF-κB Activation: A Review of the Molecular Mechanisms

**DOI:** 10.3390/ijms221910701

**Published:** 2021-10-02

**Authors:** Jan Korbecki, Donata Simińska, Magdalena Gąssowska-Dobrowolska, Joanna Listos, Izabela Gutowska, Dariusz Chlubek, Irena Baranowska-Bosiacka

**Affiliations:** 1Department of Biochemistry and Medical Chemistry, Pomeranian Medical University in Szczecin, Powstańców Wielkopolskich 72 Av., 70-111 Szczecin, Poland; jan.korbecki@onet.eu (J.K.); d.siminska391@gmail.com (D.S.); gutowska@pum.edu.pl (I.G.); dchlubek@pum.edu.pl (D.C.); 2Department of Cellular Signalling, Mossakowski Medical Research Institute, Polish Academy of Sciences, Pawińskiego 5, 02-106 Warsaw, Poland; magy80@gmail.com; 3Department of Pharmacology and Pharmacodynamics, Medical University of Lublin, Chodźki 4a St., 20-093 Lublin, Poland; joannalistos@umlub.pl

**Keywords:** cycling hypoxia, hypoxia-inducible factor, low-grade inflammation, tumor, cancer, NF-κB, HIF-1α, HIF-1β

## Abstract

Chronic (continuous, non-interrupted) hypoxia and cycling (intermittent, transient) hypoxia are two types of hypoxia occurring in malignant tumors. They are both associated with the activation of hypoxia-inducible factor-1 (HIF-1) and nuclear factor κB (NF-κB), which induce changes in gene expression. This paper discusses in detail the mechanisms of activation of these two transcription factors in chronic and cycling hypoxia and the crosstalk between both signaling pathways. In particular, it focuses on the importance of reactive oxygen species (ROS), reactive nitrogen species (RNS) together with nitric oxide synthase, acetylation of HIF-1, and the action of MAPK cascades. The paper also discusses the importance of hypoxia in the formation of chronic low-grade inflammation in cancerous tumors. Finally, we discuss the effects of cycling hypoxia on the tumor microenvironment, in particular on the expression of VEGF-A, CCL2/MCP-1, CXCL1/GRO-α, CXCL8/IL-8, and COX-2 together with PGE_2_. These factors induce angiogenesis and recruit various cells into the tumor niche, including neutrophils and monocytes which, in the tumor, are transformed into tumor-associated neutrophils (TAN) and tumor-associated macrophages (TAM) that participate in tumorigenesis.

## 1. Introduction

The growing knowledge of tumors indicates the significance of the tumor microenvironment, a collection of factors that act on cancer cells in the tumor. These factors include tumor-associated cells [1,2] along with elements of intercellular signaling, such as growth factors [3], lipid mediators [4], chemokines [5], and many others. Nutrient levels, lactic acid concentration, and acidification of the tumor microenvironment [6], as well as reduced oxygen levels, i.e., hypoxia, associated with tumor growth, are also important for tumor growth. Hypoxic conditions significantly alter the functioning of cancer cells as well as tumor-associated cells.

An important aspect of hypoxia in the tumor microenvironment is chronic low-grade inflammation. The role of inflammation supports the fight of the immune system against pathogens and is an element strengthening the anti-tumor response [7]. However, inflammatory processes also include mechanisms that inhibit the body from unduly responding to pro-inflammatory factors. They protect tissues from damage by their own over-reactive mechanisms designed to fight pathogens. During chronic inflammation, these mechanisms lead to the inhibition of anti-tumor response [8] and thus promote cancerous tumor growth [9].

This review expands on the mechanisms of the activation of hypoxia-inducible factors (HIFs) and nuclear factor κB (NF-κB) presented in our previous reviews on the effects of hypoxia on the CC [10] and CXC [11] sub-family chemokine systems. These papers show the exact mechanisms responsible for the induction of the expression of individual chemokines by chronic and cycling hypoxia. In this paper, we focus on the activation of HIFs and NF-κB by different types of hypoxia and the crosstalk between the activation pathways of these two transcription factors.

## 2. Chronic Hypoxia

### 2.1. Activation of the Hypoxia-Inducible Factor by Oxygen Reduction: The Role of Hydroxylation

The intense division of cancer cells results in the proliferation of tumor tissue. This process does not go hand in hand with angiogenesis, i.e., the formation of new blood vessels. In this way, due to the low availability of blood vessels, the tumor has areas with chronically reduced oxygen concentration. This microenvironment is called chronic (continuous, non-interrupted) hypoxia.

The most important and best-known proteins activated in hypoxia are three hypoxia-inducible factors (HIF-1, HIF-2, and HIF-3). The first two, HIF-1 and HIF-2, are responsible for the transcription of genes induced by hypoxia, while HIF-3, in addition to inducing gene expression, also decreases the activity of HIF-1 and HIF-2 [12,13,14].

All three HIFs are composed of two subunits, alpha and beta. The HIF-β subunits, also known as aryl hydrocarbon nuclear translocators (ARNT), are not regulated by any changes in oxygen, although a study on high-risk multiple myeloma cells shows that chronic hypoxia increases HIF-1β expression via NF-κB [15]. The highest expression of HIF-2β occurs in the brain and kidneys [16]. HIF-2β interferes with the function of HIF-1 and is, therefore, a suppressor gene in cancers such as oral squamous cell carcinoma [17], non-small cell lung cancer [18] and hepatocellular carcinoma [19].

In contrast to HIF-β subunits, the expression levels of HIF-1α, HIF-2α, and HIF-3α subunits are tightly regulated by changes in oxygen concentration through proteolytic degradation and transcriptional regulation. In addition, HIF-3α expression is upregulated by HIF-1 and HIF-2 [14]. This represents one of the many mechanisms of self-regulation of HIF transcriptional activity.

In normoxia, HIF-α undergoes hydroxylation on the proline residue in the N-terminal oxygen-dependent degradation domain (NODD) and C-terminal oxygen-dependent degradation domain (CODD) by three isoforms of prolyl hydroxylase (PHD) [20,21]—oxygen-dependent enzymes with an iron atom in the catalytic center [22]. PHD2 and PHD3 have similar rates of catalysis, while PHD1 has a three times lower rate than the remaining two PHDs [23].

PHDs induce the hydroxylation of the proline residues Pro^402^ HIF-1α, Pro^564^ HIF-1α, Pro^405^ HIF-2α, and Pro^531^ HIF-2α [24]. This leads to the ubiquitination of the hydroxylated HIF-α subunits by the von Hippel–Lindau protein (pVHL) [22,25,26,27], followed by the proteasomal degradation of HIF-1α and HIF-2α by 26S proteasome [28,29]. In the absence of HIF-1α and HIF-2α in the cell, active transcriptional complexes with HIF-1β and HIF-2β are not assembled.

Another factor involved in the regulation of HIF-α transcriptional activity is the factor inhibiting HIF (FIH), an oxygen-dependent enzyme with asparaginyl hydroxylase activity for HIF-α subunits. This enzyme causes hydroxylation of HIF-α at the Asn^803^ HIF-1α and Asn^847^ HIF-2α residues [30,31]. This hydroxylation inhibits the interaction of the HIF-α subunit with CBP/p300 [30,32,33]. The interaction of HIF-α with this coactivator is necessary for the transcription of HIF-dependent genes. Therefore, FIH action provides a mechanism for reducing the transcriptional activity of HIFs in normoxia.

FIH and PHD require different levels of oxygen to fully function. The activity of PHD is significantly reduced when the oxygen concentration in the cell’s environment is reduced to 5% [23]. The Michaelis constant (K_m_) for these enzymes relative to substrate oxygen is 230–250 μM [34]. In hypoxia, there is an increase in PHD expression by HIFs which increases the activity of these enzymes during reoxygenation [35,36,37,38]. In contrast, FIH requires half the oxygen concentration necessary for PHD activity [39]. In this way, FIHs inhibit the transcriptional activity of HIFs at oxygen concentrations where PHD activity is already reduced.

The reduction in PHD’s activity results in a decrease in the level of hydroxylation of the proline residue on HIF-α. This leads to (1) a decrease in the degradation of HIF-α, (2) the accumulation of these proteins in the cell, (3) the dimerization of the corresponding HIF-α and HIF-β subunits, and finally, (4) production of HIF-1, HIF-2, and HIF-3 transported to the cell nucleus. The hydroxylation of HIF-α by FIH does not occur at low oxygen concentrations. This leads to an interaction of the HIF-α subunit with CBP/p300 on the promoters of genes with hypoxia response element (HRE) sequences [30,32,33] and then to the increased expression of hypoxia-dependent genes.

The accumulation of individual HIF-α—and so, the activation of individual HIFs—depends on the duration of hypoxia [40]. HIF-1 is activated in the first 4 h of chronic hypoxia, after which HIF-1α protein levels decrease [40,41]. In contrast, maximum HIF-2α and HIF-3α levels occur after 24–48 h of hypoxia [40]. This is associated with an increased expression of hypoxia-associated factor (HAF), which causes pVHL-independent proteolytic degradation of HIF-1α [41]. In prolonged chronic hypoxia, reduced HIF-1α expression may also be caused by the activity of heat shock protein 70 (Hsp70), which, together with the carboxyl terminus of Hsc70-interacting protein (CHIP), causes the ubiquitination of HIF-1α but not HIF-2α [42].

It should be noted that the expression of HIF-2α varies in different tumors. It is absent in small cell lung carcinoma, while it is present in non-small cell lung carcinoma [43]. Additionally, an in vivo study shows that in tumor cells, the levels of HIF-1α and HIF-2α are high on average but vary depending on the type of cells [44]. In tumor-associated macrophages (TAM), HIF-2α [44,45] and HIF-1α [46] levels are high.

### 2.2. Acetylation of HIF-α as a Possible Mechanism for the Regulation of HIF’s Activity in Chronic Hypoxia

Hydroxylation is not the only mechanism that can alter HIF-α stability. Another post-translational modification that regulates the stability of HIF-α is acetylation. HIF-1α has 12 amino acid residues that are potentially subject to ubiquitination [47]. Depending on which of these residues is acetylated, this post-translational modification may either increase or decrease the stability and transcriptional activity of this HIF-1 subunit. This process has been thoroughly described for the chronic hypoxia model. Nevertheless, the effect of acetylation on HIF-1 activity in the model of cycling hypoxia is poorly understood.

In chronic hypoxia, protein 14-3-3ζ promotes the interaction of histone deacetylase (HDAC)4 with HIF-1α, which reduces the acetylation of this HIF-1 subunit [48]. As a consequence, the stability of the HIF-1α protein is increased. This mechanism has been demonstrated in a hepatocellular carcinoma model [48]. Increased stability and transcriptional activities of HIF-1α have also been observed in HDAC1, HDAC3 [49], HDAC4 [50,51,52], HDAC5 [51], and HDAC6 [50]. Importantly, HDAC4 and HDAC5 bind to HIF-1α, which prevents the hydroxylation of this subunit of HIF-1 via FIH [51]. In chronic hypoxia, HDAC7 forms a complex with HIF-1α in the cell nucleus, which increases the transcriptional activity of HIF-1 [53]. In contrast, acetylation of Lys^532^ HIF-1α by arrest defective 1 (ARD1) reduces the stability and transcriptional activity of HIF-1α [54]. In hypoxia, ARD1 expression is decreased, which increases HIF-1 activation.

Acetylation of HIF-1α may also increase the stability and transcriptional activity of this HIF subunit in chronic hypoxia. HIF-1α, but not HIF-2α, undergoes acetylation at the Lys^709^ residue by p300, which increases the stability of HIF-1α [55]. HIF-1α has 12 residues that undergo ubiquitination [47]. One of these is the Lys^709^ residue. Acetylation of this residue prevents its ubiquitination; this results in an increase in the stability of HIF-1α. In addition, the deacetylation of the Lys^674^ residue of HIF-1α in normoxia by sirtuin (SIRT)1 blocks recruitment of this HIF-1 subunit from p300 [56]. In hypoxia, a decrease in SIRT1 activity causes acetylation of Lys^674^ HIF-1α by p300/CBP-associated factor (PCAF). Other sirtuins also reduce HIF-1 pathway activation, including SIRT2 [57], SIRT3 [58], and SIRT7 [59].

### 2.3. The Role of ROS and NO in the Activation of HIFs during Chronic Hypoxia

An important part of the cellular response to hypoxia are reactive oxygen species (ROS), which increase HIF-1 stability. In chronic hypoxia, this process is much less important than the effects of ROS on signaling pathways in cycling hypoxia. Chronic hypoxia is associated with an increase in ROS generation by complex III of the mitochondrial electron transport chain [60,61,62]. Circadian locomotor output cycle protein kaput (CLOCK) may also be responsible for increasing ROS levels in chronic hypoxia [63]. ROS increase the activation of HIF-1 and NF-κB. Specifically, in the cytoplasm, ROS inhibit the activity of PHD [62,64] and FIH [31]. Importantly, the changes that ROS cause in these enzymes vary. FIH is more sensitive to ROS but is inactivated more permanently than PHD [31]. In the case of PHD, it has been suggested that ROS cause oxidation of the iron atom, important in the activity of these enzymes that regulate HIF-α stability and function [65]; no mechanism has been established for FIH [31]. At the same time, ROS activates NF-κB through various mechanisms [63,66,67], and subsequently, NF-κB increases the expression of HIF-1α mRNA. In chronic hypoxia, there is also an increase in ROS generation in cytoplasm. HIF-1 can increase NADPH oxidase (NOX)4 expression and thus ROS generation in the cytoplasm [68,69], although it is possible that, in hypoxia, NOX4 expression is increased directly by NF-κB p65/RelA [70]. Significantly, the relevance of mitochondrial ROS for HIF-1α stability is disputed by some researchers [71]. The inhibition of PHD by ROS during chronic hypoxia is also questioned [71].

Activation of HIFs is dependent on nitric oxide (NO) levels (Figure 1). This is important because many cancers are accompanied by an increase in inducible nitric oxide synthase (iNOS) expression and an increase in NO production, resulting in a poorer prognosis for the patient [72,73,74]. HIF-1α is S-nitrosylated on Cys^533^ by NO, resulting in increased stability of this protein in normoxia [75]. NO also binds to the iron atom at the catalytic center in PHD, which inhibits the activity of this enzyme [76]. On the other hand, in hypoxia, NO can interfere with HIF-1 activation and function [77]. In combination with ROS, NO increases the concentration of calcium ions in the cytoplasm which activates calpain [78]—a protease that degrades HIF-1α, independently of the 26S proteasome. In chronic hypoxia, NO also restores the activity of enzymes involved in HIF-α hydroxylation [64,79] in a mechanism dependent on the interaction between NO with ROS.

### 2.4. MAPK and AP-1 Kinases in Chronic Hypoxia

During hypoxia, mitogen-activated protein kinase (MAPK) cascades are activated and play an important role in the cellular response to reduced oxygen concentration. These processes have been thoroughly researched in the chronic hypoxia model as opposed to cycling hypoxia. Nevertheless, the activation of MAPK cascades is similar in both hypoxia, and therefore, in all likelihood, the molecular mechanisms described in this section reflect HIF-1 activation in cycling hypoxia.

The activation of MAPK cascades in chronic hypoxia is dependent on an increased concentration of calcium ions or ROS [80,81]. An influx of calcium ions in hypoxia is caused by the opening of the L-type voltage gated Ca^2+^ channels [82]. ROS or calcium ions activate the p38 MAPK [61,81,82,83,84], extracellular signal-regulated kinase (ERK) MAPK [81,82,85] and c-Jun N-terminal kinase (JNK) MAPK [81,86]. This results in the activation of c-Jun and JunB, but also a decrease in the levels of c-Fos, and JunD [87]. However, it is not only that the activation of MAPK cascades activates HIFs; in a reverse process, HIF-1 may influence the activation of MAPK cascades. The increase in c-Jun expression in chronic hypoxia in mouse embryonic fibroblasts is HIF-1 dependent [88], similar to the activation of the ERK MAPK cascade [89]. c-Jun and JunB are elements of activating protein-1 (AP-1), a transcription factor that plays an important role in the expression of various genes in hypoxia.

MAPK cascades can also affect the activation and transcriptional activity of HIF-1 (Figure 2). Chronic hypoxia induces the activation of ERK MAPK [85], p38 MAPK [90], and JNK MAPK [91]. ERK MAPK causes phosphorylation on Ser^641^ and Ser^643^ of HIF-1α. This post-translational modification of HIF-1α is important for the accumulation of this subunit in the cell nucleus and interaction of this subunit with the p300 coactivator [92,93]. Additionally, HIF-1α is phosphorylated by p38 MAPK, which increases the stability of this HIF-1 subunit [90]. Another mechanism affecting HIF-1 activation is the phosphorylation of the seven in absentia homolog 2 (SIAH2) at the Thr^24^ and Ser^29^ residues by p38 MAPK [94]. This is followed by the degradation of PHD3, an enzyme responsible for hydroxylation and degradation of HIF-1α. Activation of p38 MAPK results in decreased degradation of HIF-1α.

Chronic hypoxia increases the expression of dual specificity protein phosphatase-(DUSP)-1/mitogen-activated protein kinase phosphatase 1 (MKP1) [95]. The increased expression of DUSP1 in neurons in chronic hypoxia is dependent on neuronal nitric oxide synthase (nNOS) and NO produced by nNOS [96]. NO inactivates DUSP1; in contrast, protein kinase C (PKC)ζ is responsible for increasing DUSP1 expression in fibroblasts under chronic hypoxia [97]. However, DUSP1 expression in chronic hypoxia may also be dependent on p38 MAPK, as shown by experiments on pheochromocytoma cells [98]. The use of cobalt chloride or deferoxamine (both these compounds being PHD inhibitors) showed that HIF activation may result in increased DUSP1 expression [98], although, to date, there has been no proof of a direct effect of HIF on the expression of DUSP1. Cobalt chloride and deferoxamine may also inhibit the activity of histone demethylase which leads to an altered expression of many genes [99,100]. In chronic hypoxia, DUSP2 expression is also decreased by HIF-1 [101,102]. DUSP1 and DUSP2 are enzymes that inactivate MAPK kinases (ERK MAPK and p38 MAPK) by their dephosphorylation [103]. The upregulation of DUSP1/MKP1 expression is a mechanism that protects against excessive HIF-1 activation by ERK MAPKs in chronic hypoxia.

MAPK cascades enhance the stability and transcriptional activity of HIF-1α in hypoxia. Similar mechanisms also occur in tumor cells in normoxia, where the activation of MAPK cascades also occurs [104], in particular as a result of exposure to various growth factors [105,106].

MAPK kinases in hypoxia also cause NF-κB activation. However, which MAPK cascade is responsible for this varies from model to model. In macrophages, NF-κB activation is induced by ERK cascade [107], while in Hey-A8 human ovarian carcinoma cells, it is by p38 MAPK cascade [83]. In addition, ERK MAPK causes phosphorylation of Ser^276^ p65/RelA NF-κB which results in the activation of this transcription factor [89].

### 2.5. NF-κB Activation during Chronic Hypoxia Is Important for the Full Activation of HIFs

In hypoxia, the transcriptional activity of HIFs is increased by decreasing hydroxylation and degradation of HIF-α. However, maximal HIF activation requires the activation of other pathways. In particular, NF-κB is activated during the first hours of chronic hypoxia [40]. This leads to an increase in HIF-1α mRNA expression due to the presence of an NF-κB binding site in the *HIF1A* gene promoter [66,108,109,110,111]. Therefore, the activation of p50 NF-κB and p65/RelA NF-κB, but not c-Rel NF-κB, results in increased HIF-1α mRNA expression [112,113,114]. HIF-1β expression is also directly upregulated by NF-κB in chronic hypoxia [15,115]. The relationship of these two transcription factors is important in hypoxia because NF-κB is activated by low oxygen concentration via multiple mechanisms [116]. Like HIF-1α, the IκB kinase β subunit (IKKβ) is also hydroxylated on Pro^191^ by PHD1 [117,118]. This leads to ubiquitination of the Lys^63^ residue of IKKβ by pVHL [119]. This post-translational modification prevents transforming growth factor (TGF)-β-activated kinase 1 (TAK1) from attaching to IKKβ. This decreases IKKβ activity and thus reduces NF-κB activation. However, ubiquitination of IKKβ does not lead to proteolytic degradation of this protein [119]. In chronic hypoxia, there is a decrease in PHD1 activity, which results in a decrease in IKKβ hydroxylation by this enzyme and, consequently, in an increase in IKKβ activity. PHD2 also plays an important role in regulating the NF-κB activation pathway [120]. This enzyme indirectly regulates phosphorylation of the inhibitor of NF-κB α subunit (IκBα). Additionally, in chronic hypoxia, there is an increase in calcium ion levels in cytoplasm, which results in activation of calcium/calmodulin-dependent kinase 2 (CaMK2). This leads to ubiquitination of Lys^63^ Nemo/IKKγ by ubiquitin-conjugating enzyme 13 (Ubc13) [121]. This results in the activation of IKK.

Chronic hypoxia also affects other components of the NF-κB activation pathway. FIH can catalyze the hydroxylation of IκBα (Figure 3) [122]. Nevertheless, this has no effect and is not relevant for NF-κB activation in chronic hypoxia. Activation of CaMK2 causes Lys^21^ IκBα sumoylation by Sumo-2/3 and consequently prevents IκBα ubiquitination [121]. This leads to the release and activation of NF-κB in the absence of IκBα degradation [121]. The MAPK cascades are also activated in hypoxia, resulting in NF-κB activation. Depending on the model, ERK MAPK, which phosphorylates Ser^276^ p65/RelA NF-κB, is responsible for NF-κB activation in macrophages and primary mouse keratinocytes [89,107]. In contrast, in ovarian carcinoma cell line Hey-A8 [83] and lung adenocarcinoma A549 cells [123], the p38 MAPK cascade is responsible for NF-κB activation. In addition to these mechanisms, NF-κB activation occurs in hypoxia via the phosphatidylinositol 3-kinase (PI3K)→Akt/protein kinase B (PKB) pathway [83]. This pathway is activated either by ROS or by activation of membrane receptors [83]. Additionally, this pathway can be activated in hypoxia by 14-3-3ζ, as shown in hepatocellular carcinoma [124]. Akt/PKB kinase phosphorylates p65/RelA NF-κB cause activation of this transcription factor, but this process is independent of IκBα degradation [83]. Akt/PKB can also activate mTOR, which phosphorylates Thr^23^ IKKα and Thr^559^ and Ser^634^ IKKβ [125]. This induces IKK activation, which leads to IκBα phosphorylation and degradation. Another pathway of NF-κB activation in chronic hypoxia is the activation of protein kinase D2 (PRKD2) [126], although the exact mechanism of activation of this kinase has not been thoroughly researched.

After activation, NF-κB forms a complex with its coactivators. These complexes include PHD2 [127] and PHD3 [128]. The property of these PHDs does not depend on enzymatic activity. This is important in inflammatory reactions and in a cell’s response to chronic hypoxia. In hypoxia, PHD expression is increased in a HIF-dependent manner [35,36,37,38], which may increase the transcriptional activity of NF-κB and hence increase HIF-1α expression.

HIF-1 and HIF-2 also increase the expression of p65/RelA NF-κB in macrophages [129]. HIF-1 can also activate NF-κB indirectly by increasing the expression of alarmin receptors [130] activated by damage-associated molecular patterns (DAMPs), i.e., molecules secreted from cells during necrosis, e.g., in an environment with hypoxia. Activation of alarmin receptors results in the activation of NF-κB. There is also an increased expression of thioredoxin reductase 1 (TrxR1) [131]. This increases the level of ROS in the cytoplasm, which leads to increased activation of NF-κB [132,133].

### 2.6. Inhibition of Inflammatory Responses by Chronic Hypoxia

Hypoxia is associated with a decrease in PHD1 activity, which leads to a decrease in the hydroxylation of IKKβ [117,118] and HIF-α [20,21]. This activates this kinase which leads to NF-κB activation and an increase in HIF-1α mRNA expression by NF-κB. This is followed by transcription of hypoxia-induced genes but also some pro-inflammatory genes [40]. However, also in inflammatory responses, including the action of lipopolysaccharide (LPS), PHD is inactivated, and thereby, NF-κB is activated, which induces expression of pro-inflammatory genes and increases HIF-1α expression [134,135]. However, both pathways, i.e., NF-κB and HIF, if activated simultaneously, will be mutually exclusive (Figure 4) [136,137,138]. This is because the transcription of genes induced by hypoxia (HIF) and inflammation (NF-κB) requires the coactivator p300 [139,140] and both transcription factors compete with each other for this coactivator. Additionally, there are mechanisms of inhibition of the NF-κB pathway by the HIF activation pathway, which is important in reducing overly intense inflammatory responses as well as reducing inflammatory responses in chronic hypoxia [137,138]. TAK1 and cyclin-dependent kinase 6 (CDK6) play essential roles in this process, although the exact mechanism remains to be investigated [137]. Nevertheless, in chronic hypoxia there is a HIF-1-independent increase in IκBα expression [138]. Additionally, IκBα inhibits HIF-1α hydroxylation by FIH [141], i.e., an inhibitor of the NF-κB pathway activates the HIFs activation pathway. In inflammatory reactions, proteolytic degradation of IκBα occurs, which inhibits HIF activation by increasing FIH activity [141]. Another mechanism that reduces inflammatory responses is the increase in PHD3 expression, which blocks the interaction of IKKβ and heat shock protein 90 (Hsp90), preventing the activation of this kinase [35,36,37,142]. On the other hand, Hsp90 induces activation of PRKD2 in chronic hypoxia, which activates NF-κB [126]. The two discussed pathways may interact via other mechanisms. For example, NF-κB increases in PHD3 expression [128]. This protein, independent of its enzymatic properties, is a coactivator of p65 NF-κB [128], similar to PHD2 [127]. PHD2 and PHD3 also reduce HIF activity by participating in the degradation of HIF-α subunits. Another mechanism is the involvement of HIF-1β in TNF receptor associated factor 6 (TRAF6) expression and therefore in NF-κB activation [143]. HIF-1α decreases TRAF6 expression, probably by binding HIF-1β into the HIF-1 complex.

Thanks to the aforementioned mechanisms, there is no simultaneous response of the cell to hypoxia and to pro-inflammatory factors. Nevertheless, NF-κB is activated in chronic hypoxia, leading to an increase in the expression of some inflammatory genes [40].

### 2.7. Chronic Hypoxia vs. Cycling Hypoxia in a Tumor

The signaling pathways activated during chronic hypoxia are very well understood. Hydroxylation of HIF-α is reduced, which results in an accumulation of these subunits in the cell. Phosphorylation by MAPK kinase, change in acetylation, or influence of ROS are also responsible for the increase in HIF-α stability during chronic hypoxia. There is also an activation of NF-κB, which increases the expression of HIF-1α. Ultimately, chronic hypoxia occurs in 23 to 54% of the tumor area, depending on the tumor model and the adoption of threshold oxygen levels from which hypoxia is defined [144,145]. In comparison, cycling hypoxia covers between 29 and 62% of the tumor area, depending on the tumor model and the adopted oxygen threshold level in the definition of hypoxia [144,145]. Nevertheless, this microenvironment is not often studied. For this reason, the activated signaling pathways and thus the cellular response to cycling hypoxia are poorly understood.

## 3. Cycling Hypoxia

### 3.1. Cycling Hypoxia in a Tumor

In the initial stages of tumor growth, the intense proliferation of tumor cells is not matched by the development of blood vessels that supply cells inside the tumor with nutrients and oxygen. Therefore, chronic (continuous, non-interrupted) hypoxia occurs inside the tumor [146]. This causes activation of signaling pathways that result in angiogenesis. The blood vessels produced in the tumor are characterized by structural abnormalities [147,148]. They do not show a conventional hierarchy compared to normal vessels, which impedes blood flow. In addition, endothelial cells and pericytes are poorly connected to each other, resulting in the leakiness of blood vessels in a tumor [149]. The structural abnormalities of blood vessels result in periodic oxygen deficiencies coupled with reoxygenation in various regions of the tumor [150]. This process is known as cycling (intermittent, transient) hypoxia. This is associated with changes in the vascular blood flow pathway characterized by the absence of conventional hierarchy. There is a segmentation of the tumor into regions that experience hypoxia and normoxia at a specific time [151]. Fluctuations in oxygen concentration range from a few minutes [144,145,151,152,153] to a few hours [154,155]. At the same time, the pattern of fluctuation depends on the type of tumor, including the line that produced the tumor in in vivo studies [151,154,156]. It has been shown that the more frequent the fluctuations in oxygen levels, the stronger the responses of the cells [157,158]. In contrast, the amplitude depends on the size of the tumor. The larger the tumor, the greater the fluctuations in oxygen concentration [156].

Cycling hypoxia is a characteristic feature of malignant tumors. This type of hypoxia is also associated with further tumor growth. In particular, cycling hypoxia increases the tumor growth rate [159,160]. It also causes apoptotic resistance by increasing B-cell lymphoma-extra large (Bcl-x_L_) expression in cancer cells [161]. Simultaneously, cycling hypoxia causes migration and metastases [162,163,164] associated with the induction of the epithelial-to-mesenchymal transition (EMT) of tumor cells. Additionally, cycling hypoxia increases self-renewal of cancer stem cells [163,164], which is associated with an increased expression of transcription factor BTB and CNC homology 1 (Bach1) [160]. In experiments on macrophages, cycling hypoxia increased the pro-inflammatory phenotype of M1 macrophages, while it had no pro-inflammatory effect on M2 macrophages [165]. On the other hand, experiments on lung carcinoma LLC1 cells have shown that cycling hypoxia reduces the number of M1 macrophages in a tumor [159]. In contrast, chronic hypoxia causes M2 polarization of macrophages [166].

### 3.2. Cycling Hypoxia: Intracellular Signaling Pathways

Cycling hypoxia alters the expression of fewer genes than chronic hypoxia [167], although the cellular response is similar for both. Cycling hypoxia has been shown to strongly activate the epidermal growth factor (EGF) pathway through a greater (compared to chronic hypoxia) increase in the expression of activators of the epidermal growth factor receptor (EGFR) family of receptors [167]. Cycling hypoxia, just like chronic hypoxia, is pro-inflammatory [168,169]. By activating NF-κB, it increases the expression of pro-inflammatory genes including cyclooxygenase-2 (COX-2).

Both discussed types of hypoxia alter the expression of similar genes, due to the fact that cycling hypoxia activates HIF-1 and NF-κB, the same transcription factors as chronic hypoxia [167,170]. However, the mechanisms of activation, as well as the degree of their activation, are different. In cycling hypoxia, activation of HIF-1 is stronger and longer [163,171,172]. In addition, the expression level of HIF-1α protein is increasingly higher with successive hypoxia cycles [173]. The higher the frequency of cycles (number of cycles per hour), the higher the activation of HIFs [158]. In contrast, reoxygenation is followed by HIF-α degradation, in part through increased PHD expression via HIF-1 activated in the period of reduced oxygen concentration [35,36,37,38]. The frequency of oxygen concentration fluctuations in the tumor depends on the cell line that produced the tumor [151,154,156]. Additionally, the more frequent the fluctuations in oxygen concentration, the greater the activation of NF-κB [157,158].

The mechanism of HIF-α accumulation in cycling hypoxia is ROS-dependent [161,174]. Cycling hypoxia induces the upregulation of NOX1 [173] and NOX4 [172,175], which generate ROS. ROS is also produced via the activation of xanthine oxidase [176,177], which may occur due to a NOX2-induced increase in intracellular calcium levels [177]. Another source of ROS in cycling hypoxia is the mitochondrial electron transport chain [178,179]. Increased levels of ROS increase the synthesis and stability of HIF-1α by decreasing PHD activity [171,180], probably due to the oxidation of the iron atom which is important in PHD activity [62,64,65]. ROS also inactivates FIH, although the exact mechanism of this inactivation has been poorly researched [31]. On the other hand, ROS causes activation of calpains, which cause HIF-2α degradation in cycling hypoxia [176]. Nevertheless, more research is required on whether HIF-2 has some role in cycling hypoxia.

Other signaling pathways also play an important role in cycling hypoxia. In particular, ROS activates MAPK cascades—ERK MAPK [175]—and the activation of JNK MAPK cascade leads to the activation of AP-1 [178]. However, there are no studies showing the effect of activated MAPK cascades in cycling hypoxia on phosphorylation of the HIF-1α subunit. This subunit may undergo phosphorylation in chronic hypoxia, which increases its stability and accumulation in the cell nucleus [92,95,181,182]. In cycling hypoxia, there is increased expression of DUSP1 [183,184]—a phosphatase that inactivates MAP kinases [103] but also increases the expression of manganese superoxide dismutase (MnSOD) that decreases ROS levels [184]. ROS activate phospholipase C-γ (PLCγ), resulting in increased calcium ion levels in the cytoplasm and PKC activation [177,180]. This leads to the activation of the mammalian target of rapamycin (mTOR). This kinase, probably through S6K, causes phosphorylation of HIF-1α and thus increases its stability.

Cycling hypoxia also causes activation of protein kinase A (PKA) (Figure 5) [185]. This kinase causes the phosphorylation of Thr^63^ and Ser^692^ in HIF-1α, which increases the stability of this HIF-1 subunit [185,186]. The mechanism of PKA activation is independent of cyclic adenosine monophosphate (cAMP) but is ROS-dependent [187]. The increase in HIF-1α stability induced by phosphorylation by PKA and mTOR is independent of PHD and oxygen levels.

In cycling hypoxia, ROS activates nuclear factor erythroid 2-related factor (Nrf2) [173], which results in an increased expression of thioredoxin 1 (Trx1), which then increases HIF-1α signaling. This effect is related to the interaction of Trx1 with HIF-1 in the cell nucleus, but not to the reductase activity of Trx1 [188,189].

Cycling hypoxia also causes changes in HIF-1α acetylation. Cycling hypoxia results in decreased expression of HDAC3 and HDAC5 proteins, but not the other HDACs [190], as demonstrated in rat pheochromocytoma PC12 cells. A change in HDAC5 activity levels increases HIF-1α acetylation and thus the stability of this HIF-1 subunit. With this model, HDAC3 does not appear to affect transcriptional activity or HIF-1α stability. There are no other (i.e., other than Wang et al. 2021) studies on the alteration of HIF-1α acetylation in cycling hypoxia. However, in normoxia and chronic hypoxia, this post-translational modification of the HIF-1α subunit is frequently studied [48,49,52,53,56].

In cycling hypoxia, ROS activates NF-κB [161,174,175,191]. In breast cancer, this is dependent on IκBα degradation [169]. In melanoma, this process is independent of IKK activation and of IκBα degradation [192]. NF-κB activation may depend on ERK MAPK, p38 MAPK, and JNK MAPK [170,179,193,194]. On the other hand, a study on cycling hypoxia in the renal tubular epithelial cell model showed a reduction in the expression of ubiquitin-specific peptidase 8 (USP8) [195], which leads to ubiquitination of Lys^63^ TAK1 and activation of the entire NF-κB activation pathway.

### 3.3. Cycling Hypoxia: Effects on the Tumor Microenvironment

Chronic hypoxia is accompanied by an activation of NF-κB, which increases the expression of HIF-1α and some pro-inflammatory genes [40]. However, it seems that, in cycling hypoxia, the activation of NF-κB is greater than in chronic hypoxia, and thus this type of hypoxia has a very pro-inflammatory character [168,169,192,196]. It increases the expression of genes associated with inflammatory responses as well as increases the cellular response to pro-inflammatory factors. In particular, there is an increased expression of COX-2 [167,168,169,197], CC motif chemokine ligand 2 (CCL2)/monocyte chemoattractant protein 1 (MCP-1) [192,194,198,199], CXC motif chemokine ligand (CXCL)1/growth related oncogene (GRO)-α [167], CXCL8/interleukin (IL)-8 [167,168,200,201], and IL-6 [168]. All of these are inflammatory mediators involved in various neoplastic processes.

Both types of hypoxia also increase vascular endothelial growth factor (VEGF)-A expression. This effect depends on the cancer cell line. VEGF-A expression in the tumor cell is increased much more under chronic hypoxic conditions than in cycling hypoxia. This has been shown in melanoma WM793B cells and prostate cancer PC-3 cells [167], as well as hepatocellular carcinoma HepG2 cells [202]. At the same time, in ovarian cancer SK-OV-3 cells, cycling hypoxia did not increase VEGF-A expression [167]. However, in melanoma A-07 cells, both types of hypoxia increased VEGF-A equally [203].

VEGF-A is one of the best described pro-angiogenic factors in a tumor (Figure 6) [204,205]. However, in a tumor, there is not only VEGF-A, but also other angiogenesis-inducing factors. These include factors whose expression is associated with cycling hypoxia, in particular the aforementioned CCL2/MCP-1, CXCL1/GRO-α, CXCL8/IL-8, and prostaglandin E_2_ (PGE_2_)—the product of COX-2 activity. The main mechanism of the proangiogenic properties of CCL2/MCP-1 is the recruitment of monocytes into the tumor niche, which are transformed into TAM [206,207], which secrete VEGF-A but also other proangiogenic factors such as matrix metalloproteinase 9 (MMP-9) and PGE_2_ [208]. CCL2/MCP-1 can also directly act on endothelial cells [209]. CXCL1/GRO-α and CXCL8/IL-8 are chemokines that activate CXC motif chemokine receptor (CXCR)2 [210,211], responsible for their pro-angiogenic properties [212,213,214]. These chemokines also recruit tumor-associated neutrophils (TAN) to the tumor niche [215,216,217,218]—cells that secrete MMP-9 into the tumor microenvironment [219,220]; MMP-9 is a metalloproteinase that causes a VEGF release from the extracellular matrix (ECM) [221].

PGE_2_ is also a pro-angiogenic factor, although not directly. It participates in angiogenesis and lymphangiogenesis by increasing the expression of various angiogenic and lymphangiogenic factors such as VEGF-A, VEGF-C, basic fibroblast growth factor (bFGF), platelet-derived growth factor (PDGF), endothelin-1 [222,223,224,225,226] and causes an increase in the expression of CXCR4, the receptor for angiogenic CXCL12 [227].

The aforementioned pro-inflammatory factors induced by cycling hypoxia also act on tumor-associated cells. For example, they recruit various cells into the tumor niche. CCL2/MCP-1 is a TAM recruiting factor [206,228,229], while CXCL1/GRO-α and CXCL8/IL-8 are TAN recruiting factors [215,216,217,218]. PGE_2_, through its action on anti-tumor cells, is one of the mechanisms of cancer immunoevasion. It inhibits the anticancer function of NK cells and dendritic cells and enhances the pro-cancer function of M2 macrophages and regulatory T cells (T_reg_) [230,231,232,233,234].

## 4. Mediators of Inflammatory Responses Induced by Chronic Hypoxia as a Therapeutic Target

Cycling hypoxia is a feature of all solid tumors [145,151,152,153,154,155]. It activates the same signaling pathways and alters the tumor microenvironment identically or similarly in all tumors. Therefore, understanding the mechanisms of action of cycling hypoxia will either allow the development or improvement of anti-cancer therapies against many types of cancer.

Cycling hypoxia is associated with elevated COX-2 expression and consequently an increase in PGE_2_ production [167,168,169,197]. For this reason, the use of nonsteroidal anti-inflammatory drugs (NSAID) together with standard anticancer therapy provides beneficial effects for patients with various solid tumors [235], especially breast cancer [236], colorectal cancer [235], oesophageal cancer [235], and prostate cancer [237]. Nevertheless, in patients with non-small-cell lung cancer, NSAIDs improve the overall response rate but have no effect on patient survival after therapy [238,239,240].

Additionally, cycling hypoxia increases CCL2/MCP-1 production in the tumor [192,194,198,199]. Therefore, taking the CCL2→CC motif chemokine receptor 2 (CCR2) axis as a therapeutic target is an approach with great therapeutic potential. In particular, a CCR2 antagonist [241,242,243] and CCL2-neutralizing antibody [244,245,246,247] are being tested for the treatment of many types of cancer.

In addition to CCL2/MCP-1, cycling hypoxia increases in the expression of CXCL1/GRO-α [167] and CXCL8/IL-8 [167,168,200,201]. For this reason, a CXCL1-neutralizing antibody [248] and CXCL8-neutralizing antibody [249,250,251,252] are being tested as potential anticancer agents. Another therapeutic approach is the use of receptor antagonists for sub-family CXC chemokines, such as CXCR2 antagonists SB225002 [253,254,255] and SB265610 [256]. CXCR1/CXCR2 dual antagonists that act on both CXCL8/IL-8 receptors have also been tested [257,258,259,260,261]. Because CXCR2 is a receptor for CXCL1/GRO-α [210], such dual antagonists also reduce the effects of this chemokine. It is also possible during cancer therapy to inhibit the entire NF-κB activation pathway by using proteasome inhibitors and IKKβ inhibitors [262]. This prevents the activation of NF-κB and so the expression of all genes is dependent on the activation of this transcription factor by cycling hypoxia.

Another option is to improve the anti-cancer anti-angiogenic therapy, e.g., by using bevacizumab—an anti-VEGF-A monoclonal antibody [263]. However, resistance to bevacizumab is very common, which is related to the presence of pro-angiogenic factors other than VEGF-A in the tumor. These factors complement or, in the absence of VEGF-A, replace VEGF-A in their functions [263,264]. For this reason, it has been suggested that bevacizumab be used together with drugs that inhibit other pro-angiogenic factors, particularly those induced by cycling hypoxia, such as anti-CCL2 antibody [265] and CCR2 inhibitor [266]. CCR2 is the receptor for CCL2/MCP-1 and both therapeutic approaches target the CCL2→CCR2 axis. Another option is to use bevacizumab with NSAID [267], mainly COX-2 inhibitors that reduce PGE_2_ production. As already mentioned, PGE_2_ has no direct angiogenic effect, but it increases the expression of pro-angiogenic factors [222,223,224,225,226]. Therefore, decreased PGE_2_ production results in decreased expression of other pro-angiogenic factors. Another possibility is to combine bevacizumab with a CXCR1/CXCR2 dual inhibitor [268]. It is also possible to combine bevacizumab with an inhibitor of the NF-κB activation pathway, e.g., NPI-0052/salinosporamide A, which is a proteasome inhibitor that blocks proteolytic degradation of IκBα [269]. NF-κB activation in cycling hypoxia is the most important mechanism in increasing the expression of all the aforementioned pro-angiogenic factors [168,169,192,196]. Therefore, decreased NF-κB activation decreases the expression of pro-angiogenic factors induced by this transcription factor.

## 5. Conclusions: A Perspective for Further Research on Chronic Hypoxia

The vast majority of published in vitro experiments on hypoxia in cancer relate to chronic hypoxia. Most of the available work has not investigated the effect of cyclic changes in oxygen concentration on tumor cells. For this reason, this type of research model does not reflect the actual state of the cancerous tumor, with cycling hypoxia affecting a considerable part of the tumor. In this way, the results of studies showing the effect of chronic hypoxia only reflect the situation in one area in a tumor. For this reason, it is advisable that each study on hypoxia in a tumor should use an in vitro model that includes cycling hypoxia.

## Figures and Tables

**Figure 1 ijms-22-10701-f001:**
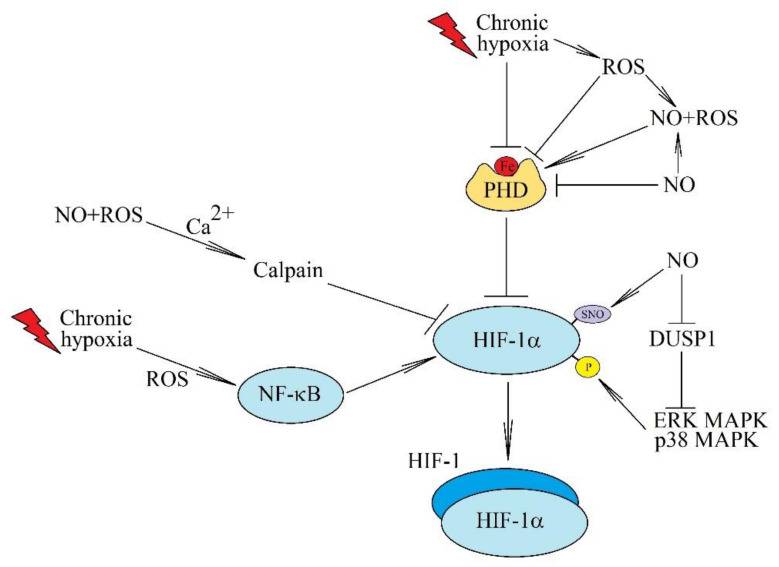
The effects of ROS and NO on the activation of HIF-1 in chronic hypoxia. Chronic hypoxia there is associated with an increase in the level of ROS which inactivate FIH and PHD. This increases the activation of HIF-1. ROS are also involved in the activation of NF-κB, a transcription factor important in the full activation of HIF-1. HIF-1 activation can also be induced by NO, especially at sites of inflammatory reactions. NO causes the S-nitrosylation of HIF-1α, which increases the stability of this protein. Another post-translational modification of HIF-1α induced by NO is phosphorylation associated with the inactivation of DUSP1. NO can also bind to the iron atom in PHDs and thus inactivate these enzymes. However, in combination with ROS, NO can restore activity of PHDs in chronic hypoxia. It can also increase calcium ion levels in the cytoplasm which activates calpain—a protease that degrades HIF-1α independently of the 26S proteasome.

**Figure 2 ijms-22-10701-f002:**
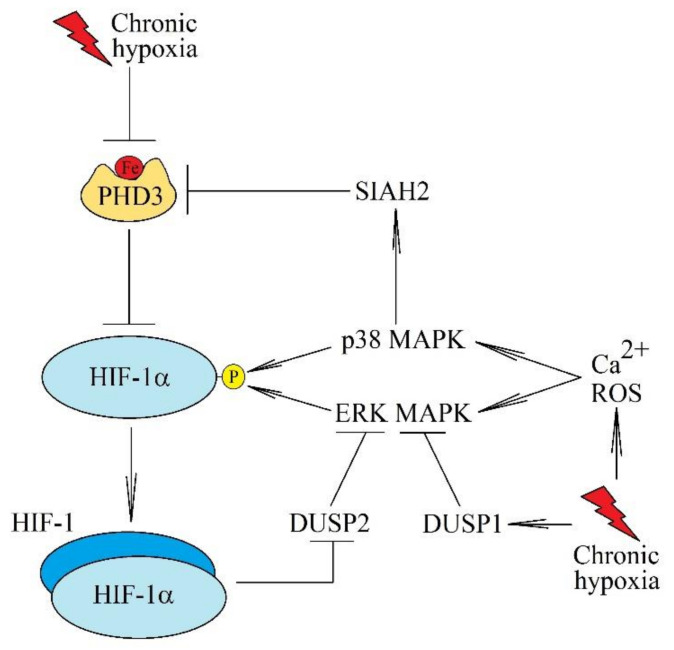
The importance of MAPK cascades for the HIF-1 activation pathway. During chronic hypoxia, MAPK cascades, in particular ERK MAPK and p38 MAPK, are activated by ROS and increased calcium ions. These kinases cause phosphorylation of HIF-1α and consequently increase the stability and transcriptional activity of HIF-1. p38 MAPK can also cause the activation of SIAH2, which results in the ubiquitination and degradation of PHD3. Important in this model of HIF-1 activation are also phosphatases, in particular DUSP1 and DUSP2—enzymes that catalyze a reaction reverse to ERK MAPK and p38 MAPK. In chronic hypoxia, there is a decrease in DUSP2 expression but an increase in DUSP1, which is a mechanism for regulating HIF-1 activation.

**Figure 3 ijms-22-10701-f003:**
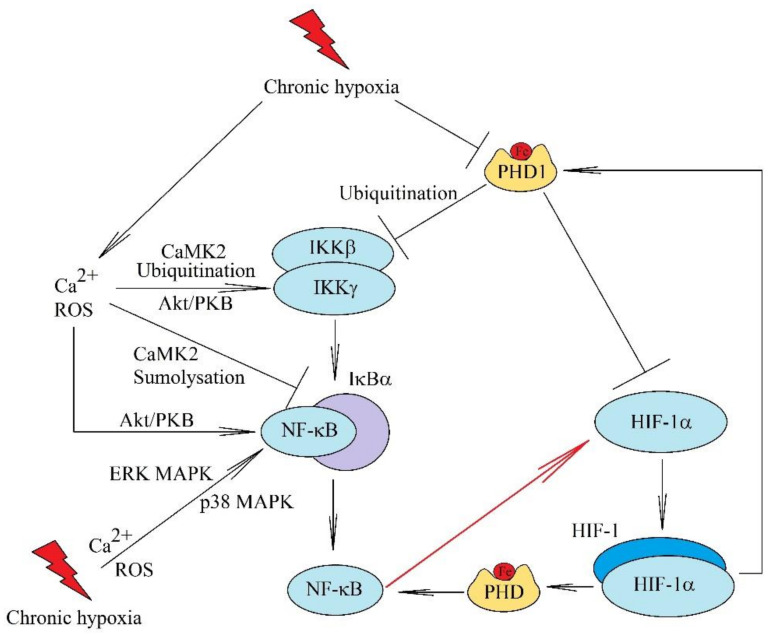
The hypoxia-induced mechanism of NF-κB activation. In hypoxia, NF-κB is an important factor in the increase in HIF-1 mRNA expression, which is activated when oxygen concentration is decreased. This process occurs through multiple pathways. Like HIF-1α, IKKβ activation is inhibited by hydroxylation by PHD1. In hypoxia, PHD1 activity is reduced, which enables IKKβ activation. NF-κB activation during hypoxia also involves ROS and calcium ion mobilization into the cytoplasm. These factors cause the ubiquitination of IKKγ/NEMO, which increases IKK activity. IκBα is SUMOylated, which decreases the activity of this inhibitor of the NF-κB activation pathway. Chronic hypoxia is also associated with the activation of kinases such as p38 MAPK, ERK MAPK, and Akt/PKB, which phosphorylate NF-κB and IKKβ, thus activating this transcription factor.

**Figure 4 ijms-22-10701-f004:**
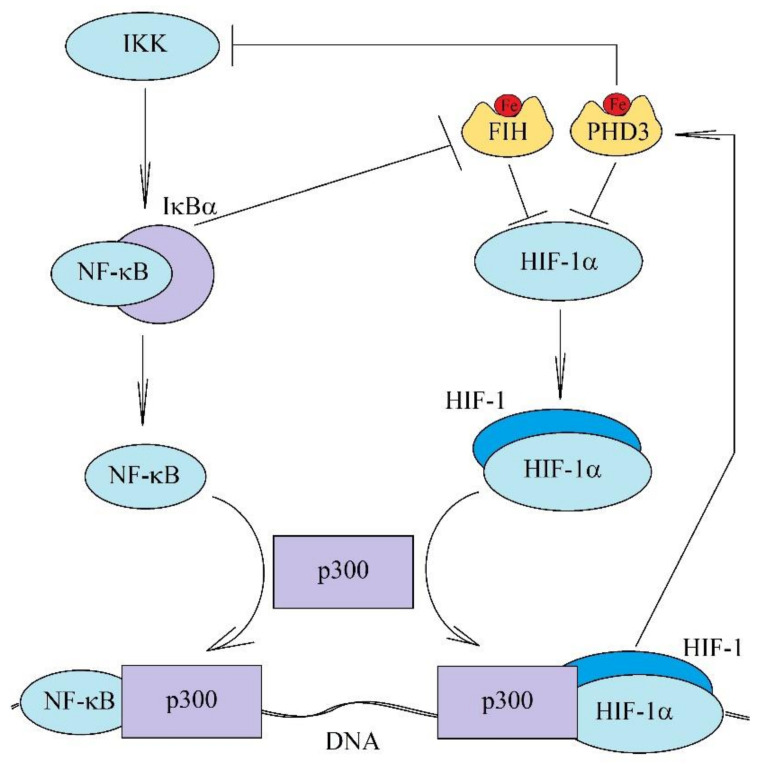
The inhibition of the NF-κB pathway activation by HIF. Chronic hypoxia is associated with NF-κB activation, although there are also mechanisms that silence the proinflammatory response, such as an increase in PHD3 expression, which inhibits IKK activity. Additionally, there is an HIF-1 induced increase in the expression of IκBα, an inhibitor of NF-κB. The simultaneous activation of NF-κB and HIF causes these two transcription factors to compete for the coactivator p300.

**Figure 5 ijms-22-10701-f005:**
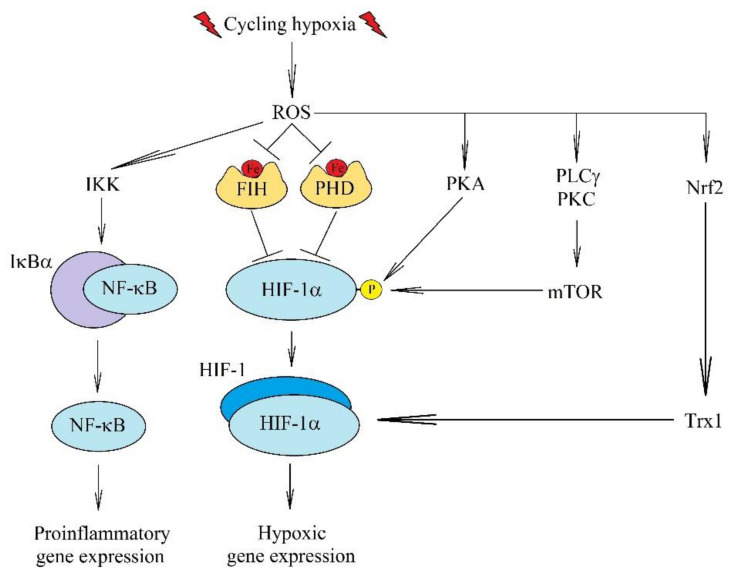
Effect of ROS in cycling hypoxia on the activation of HIF-1 and NF-κB. Cycling hypoxia induces the generation of ROS, which cause the activation of HIF-1 and NF-κB. In particular, ROS inactivate FIH and PHD, which results in increased stability of HIF-1α protein. ROS also activate PKA and mTOR, which phosphorylate HIF-1α and thus increase the stability of this protein and its accumulation in the cell. ROS also causes an increase in the expression of Trx1, which enhances the transcriptional activity of HIF-1.

**Figure 6 ijms-22-10701-f006:**
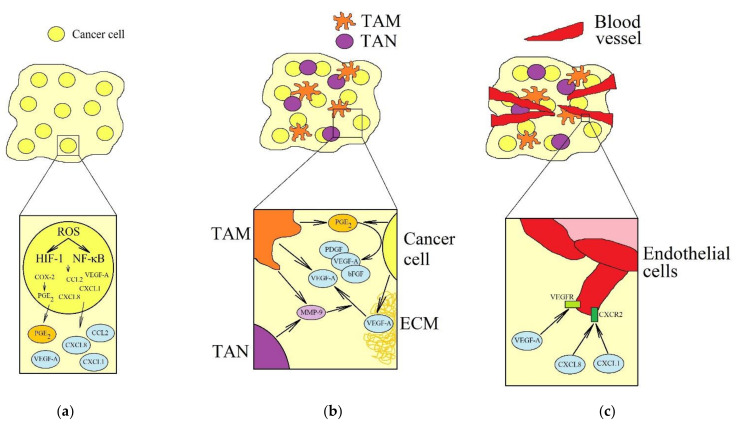
Effect of cycling hypoxia on angiogenesis in cancer. Cycling hypoxia activates HIF-1 and NF-κB in the tumor cell. (**a**) This leads to increased production of VEGF-A, CCL2/MCP-1, CXCL1/GRO-α, CXCL8/IL-8, and PGE_2_. (**b**) Subsequently, CCL2/MCP-1, CXCL1/GRO-α and CXCL8/IL-8 induce recruitment of TAM and TAN to the tumor niche. Cells that possess pro-angiogenic properties. TAN secrete MMP-9 into the tumor microenvironment, whereas TAM secrete MMP-9 and VEGF-A. MMP-9 is a metalloproteinase that releases VEGF-A. PGE_2_ also increases the expression of proangiogenic factors. (**c**) VEGF-A, CCL2/MCP-1, CXCL1/GRO-α and CXCL8/IL-8 directly cause angiogenesis.

## Data Availability

Not applicable.

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
