# Peer review of "Chronic and Cycling Hypoxia: Drivers of Cancer Chronic Inflammation through HIF-1 and NF-κB Activation: A Review of the Molecular Mechanisms"

_ijms, 2021, doi:10.3390/ijms221910701_

Round 1

Reviewer 1 Report

The review is written comprehensively, provides multilevel comparison of chronic and transient hypoxia, and in detail describes their regulatory consequences on the behaviour and signalling within a cancer cell. Although the topic of different types of hypoxia triggering similar but not identical signalling pathways and cellular processes involved in tumour growth is being discussed for several decades, clear understanding of role of cyclic hypoxia still needs deeper elucidation. Knowledge on its contribution to aggressive growth or chemoresistant phenotype of tumours might help to better understand tumour microenvironment and identify potential novel targets for more effective treatments.  

Comments for minor revision:

Lane 103 – value of Km of 230-250 mM should be checked, related citations seem to describe Km in mM

Lane 154-156 -  One of these is the Lys 709  residue, which indicates  that the acetylation of Lys 709  blocks ubiquitination is important in the proteolytic degradation  of  HIF-1α. – this sentence needs revision

Lane 192 -  Chronic hypoxia there is associated with an increase in the level of ROS which cause oxidation of the iron atom in  FIH and PHD, and so inactivation of these enzymes. – this sentence needs revision

Lane 208 – …both hypoxias… – rather both hypoxia

Lanes 382-385 and 401-403 – a sentence is duplicated

Lane 445 – HIF-a should be replaced with HIF-α

Lane 481 -...ROS also activate....

Lane 485-487 -  Cycling hypoxia results  in  decreased  expression  of  HDAC3  and  HDAC5  proteins,  but  not  the  other  HDACs  [189], as demonstrated in rat pheochromocytoma PC12 cells.   – is this true also for other type of cells? Other cell types are not discussed...

Lane 493 -...ROS also activate....

Lane 603 - The vast majority of published in vitro experiments on hypoxia in cancer only relate to chronic hypoxia. – it might be more realistic to say … The vast majority of published in vitro experiments on hypoxia in cancer mostly relate to chronic hypoxia.

Author Response

Review 1

The review is written comprehensively, provides multilevel comparison of chronic and transient hypoxia, and in detail describes their regulatory consequences on the behaviour and signalling within a cancer cell. Although the topic of different types of hypoxia triggering similar but not identical signalling pathways and cellular processes involved in tumour growth is being discussed for several decades, clear understanding of role of cyclic hypoxia still needs deeper elucidation. Knowledge on its contribution to aggressive growth or chemoresistant phenotype of tumours might help to better understand tumour microenvironment and identify potential novel targets for more effective treatments.  

Comments for minor revision:

Lane 103 – value of Km of 230-250 mM should be checked, related citations seem to describe Km in mM

It has been corrected according to the Reviewer’s comment.

Lane 154-156 -  One of these is the Lys 709  residue, which indicates  that the acetylation of Lys 709  blocks ubiquitination is important in the proteolytic degradation  of  HIF-1α. – this sentence needs revision

It has been corrected according to the Reviewer’s comment.

Lane 192 -  Chronic hypoxia there is associated with an increase in the level of ROS which cause oxidation of the iron atom in  FIH and PHD, and so inactivation of these enzymes. – this sentence needs revision

It has been revised according to the Reviewer’s comment, also in the paragraph about cycling hypoxia.

Lane 208 – …both hypoxias… – rather both hypoxia

It has been corrected according to the Reviewer’s comment.

Lanes 382-385 and 401-403 – a sentence is duplicated

One of the sentences has been removed.

Lane 445 – HIF-a should be replaced with HIF-α

It has been corrected according to the Reviewer’s comment in this and other two places.

Lane 481 -...ROS also activate....

Lane 485-487 -  Cycling hypoxia results  in  decreased  expression  of  HDAC3  and  HDAC5  proteins,  but  not  the  other  HDACs  [189], as demonstrated in rat pheochromocytoma PC12 cells.   – is this true also for other type of cells? Other cell types are not discussed...

In the available literature, changes in HDAC expression have been examined only in one cell line and that is why we do not discuss other types of cells.

Lane 493 -...ROS also activate....

Lane 603 - The vast majority of published in vitro experiments on hypoxia in cancer only relate to chronic hypoxia. – it might be more realistic to say … The vast majority of published in vitro experiments on hypoxia in cancer mostly relate to chronic hypoxia.

It has been corrected according to the Reviewer’s comment.

Reviewer 2 Report

This Review describes in detail the molecular mechanisms Hif- and NFkB-associated in cancer chronic inflammation, in addition to provide clear and, probably little known, information and related importance about cycling inflammation. 

The Review is clearly written,  interesting and original, figures are appropriate and, even more important, it points out a conditioned often underestimated by researchers in the oncologic ambit that deserve major attention.

Author Response

Review 2

This a very well written and organized review.

Much was said about HIF1alpha, but I was wondering how is the control of HIF1beta. 
There is a TF which can controls HIF1beta and also interferes with NFkappaB, the well known aryl hydrocarbon receptor. I believe it be interesting to the authors to include a comment in this subject.

We have added a fragment on the regulation of HIF-1beta by NF-kappaB.

Reviewer 3 Report

This a very well written and organized review.

Much was said about HIF1alpha, but I was wondering how is the control of HIF1beta. 
There is a TF which can controls HIF1beta and also interferes with NFkappaB, the well known aryl hydrocarbon receptor. I believe it be interesting to the authors to include a comment in this subject.

Author Response

Review 3

This Review describes in detail the molecular mechanisms Hif- and NFkB-associated in cancer chronic inflammation, in addition to provide clear and, probably little known, information and related importance about cycling inflammation. 

The Review is clearly written,  interesting and original, figures are appropriate and, even more important, it points out a conditioned often underestimated by researchers in the oncologic ambit that deserve major attention.

Thank you very much